# Tau Post-Translational Modifications: Potentiators of Selective Vulnerability in Sporadic Alzheimer’s Disease

**DOI:** 10.3390/biology10101047

**Published:** 2021-10-15

**Authors:** Trae Carroll, Sanjib Guha, Keith Nehrke, Gail V. W. Johnson

**Affiliations:** 1Department of Pathology, University of Rochester Medical Center (URMC), Rochester, NY 14642, USA; Trae_Carroll@urmc.rochester.edu; 2Department of Anesthesiology and Perioperative Medicine, University of Rochester Medical Center (URMC), Rochester, NY 14642, USA; sanjib_guha@urmc.rochester.edu; 3Department of Medicine, Nephrology Division, University of Rochester Medical Center (URMC), Rochester, NY 14642, USA; Keith_Nehrke@urmc.rochester.edu

**Keywords:** aging, Alzheimer’s disease, autophagy, neurodegeneration, selective vulnerability, tau, tau PTMs

## Abstract

**Simple Summary:**

The spatial progression of most Alzheimer’s Disease (AD) cases follows a well-defined route through the brain, yet research on the disease has yet to determine why this specific pattern of disease progression occurs. Recent developments have revealed that certain neurons are more vulnerable to AD stress than others, a concept which has since been termed “selective vulnerability”. Determining the cellular mechanisms that make certain neurons more vulnerable to disease could provide critical insights into how the disease leads to neurodegeneration and ultimately causes dementia. This review explores how modification of tau protein, a critical mediator of AD pathogenesis, may contribute to selective vulnerability and identifies key cellular components that may be involved at the earliest stages of disease progression.

**Abstract:**

Sporadic Alzheimer’s Disease (AD) is the most common form of dementia, and its severity is characterized by the progressive formation of tau neurofibrillary tangles along a well-described path through the brain. This spatial progression provides the basis for Braak staging of the pathological progression for AD. Tau protein is a necessary component of AD pathology, and recent studies have found that soluble tau species with selectively, but not extensively, modified epitopes accumulate along the path of disease progression before AD-associated insoluble aggregates form. As such, modified tau may represent a key cellular stressing agent that potentiates selective vulnerability in susceptible neurons during AD progression. Specifically, studies have found that tau phosphorylated at sites such as T181, T231, and S396 may initiate early pathological changes in tau by disrupting proper tau localization, initiating tau oligomerization, and facilitating tau accumulation and extracellular export. Thus, this review elucidates potential mechanisms through which tau post-translational modifications (PTMs) may simultaneously serve as key modulators of the spatial progression observed in AD development and as key instigators of early pathology related to neurodegeneration-relevant cellular dysfunctions.

## 1. Introduction

Alzheimer’s disease (AD) is the most common form of dementia among older adults, accounting for approximately 70% of all cases, with its incidence increasing by nearly ten million cases each year [1]. The most notable hallmarks of AD are extracellular plaques and intraneuronal neurofibrillary tangles (NFTs), aggregates comprised of amyloid-beta peptide (Aβ) and tau protein, respectively. For decades, these aggregates were thought to represent key pathological agents that mediated AD. However, more recently that dogma has been challenged. While Aβ and tau aggregates may exacerbate immune responses and tissue-level dysfunction, there is now significant evidence supporting a greater relative impact of soluble oligomers of tau on disease progression [2,3]. Moreover, while the “spotlight” in the past was predominantly on Aβ as a perpetrator of AD pathogenesis, it has become increasingly clear that tau also plays a central role in the development and progression of AD [4,5]. This viewpoint is strengthened by the fact that pathological forms of tau can independently lead to neurodegenerative disorders, collectively referred to as primary tauopathies which include conditions such as corticobasal degeneration (CBD), Pick’s Disease (PiD), progressive supranuclear palsy (PSP), and frontotemporal lobar degeneration with tau pathology (FTLD-tau) [6,7,8].

While AD is considered a secondary tauopathy, as it is also defined by the presence of Aβ pathology, tau and the factors that cause it to become pathological are clearly and critically relevant. A defining feature in the majority of sporadic AD cases is that tau pathology progresses sequentially through defined regions of the brain, and the propagation of tau pathology from one region to the next strongly correlates with disease severity [9]. This observation of a conserved temporal-spatial pathogenesis is not limited to AD but is common among neurodegenerative disorders, and it is frequently termed “selective vulnerability” [10]. Selective vulnerability in AD is regionally defined by Braak staging, a classification system used to describe the extent to which tau pathology has spread throughout a patient’s brain [11]. In this context, brain regions that develop pathology first are considered the most vulnerable to disease and those that develop pathology later are more resistant. Further study has revealed that neuronal subpopulations within the same region are also more or less vulnerable to tau pathology. Identifying the biological differences between vulnerable and resistant neurons is likely to provide a means for discerning the molecular causes behind early AD pathogenesis. Interestingly, multiple studies have shown that Aβ plaque localization does not correlate with the spatial progression of AD, whereas patterns of NFT accumulation effectively mirror the disease progression pathway [11,12,13], indicating that tau pathology is more tightly linked with AD selective vulnerability than is Aβ pathology.

This relationship between tau pathology and selective vulnerability, together with the concept that soluble tau oligomers are a primary toxic moiety, provides a framework for analyzing how toxic tau epitopes may contribute to selective vulnerability in AD. For example, the function, structural conformation, and localization of soluble tau is highly dependent on its post-translational modifications (PTMs) [14]. Despite knowing that tau’s PTMs drastically alter its physicochemical properties, how specific PTMs contribute to tissue-specific neurodegeneration has still not been clearly defined. As such, this review will primarily explore the development of tau PTMs in the context of disease progression and suggest mechanisms through which tau modifications may initiate AD pathology upstream of insoluble aggregates and NFT formation. In the model proposed within, the spatial progression of tau pathology is driven by the local accumulation of toxic tau epitopes that potentiate subsequent pathology in neighboring cells. Specifically, tau PTMs may potentiate the selective vulnerability observed in sporadic AD in three major ways: by initiating native tau’s transformation into a toxic entity, by increasing the propensity for toxic tau to accumulate in individual cells, and by promoting the propagation of toxic tau from one brain region to the next.

## 2. Tau Structure, Function, and Pathology

Human tau is a relatively hydrophilic protein that is encoded by the *MAPT* gene on chromosome 17 [15]. Originally discovered while studying factors that promote microtubule assembly [16], interest in tau spiked when it was found to be the primary component of AD-associated NFTs [17,18,19]. In the human brain, there are six major isoforms of tau which range from 352 to 441 amino acids in total size, with 0, 1, or 2 N-terminal inserts of 29 amino acids and 3 or 4 microtubule binding domains (MTBDs) of 31 or 32 amino acids (Figure 1) [20]. All isoforms lack significant secondary structure and remain mostly unstructured when present in the cytoplasm [21]. There are many excellent reviews of tau isoforms, alternate splice variants, and functional domains [15,22,23].

Classically, tau was considered to be a protein that stabilized microtubules in healthy neurons, but recent studies have challenged this supposition [25]. Other studies have shown that tau’s binding to microtubules is dynamic and transient, occurring through a “kiss and hop” mechanism [26]. This more dynamic view of tau has diversified the repertoire of tau’s involvement in cellular processes to include increasing microtubule dynamics and the length of labile domains in microtubules [27], modulating axonal transport [28], regulating synaptic plasticity [29], maintaining genomic integrity [30,31], controlling iron transport [32], and regulating mitochondrial networks [33,34,35]. Studies to date have not concretely identified how tau’s microtubule binding affects these other cellular processes; thus, it has become apparent that factors beyond microtubule stabilization need to be considered when assessing the physiologic and molecular consequences of toxic tau conversion. Decades of research have revealed many ways in which pathologically modified tau may facilitate neurodegeneration (Figure 1), yet the underlying mechanisms behind tau-induced pathology remain scattered and unclear. Toxic tau has been shown to induce inefficient transportation of intracellular cargo to distal compartments of the neuron—including mitochondria—and hinders synaptic function [23,36,37,38]. Not only can abnormal tau limit mitochondrial transport, it can also directly interfere with mitochondrial function, impacting energy bioavailability, creating reactive oxygen species, and generating proteostatic stress for the neuron [39]. Moreover, dysregulated tau can accumulate within neurons by impairing autophagic clearance, thereby triggering the endoplasmic reticulum’s unfolded protein response and inducing proteostatic stress [40,41,42]. In addition, tau may induce pathology by epigenetically modulating gene transcription [43,44]. Ultimately, it is still uncertain which of these tau-induced dysfunctions primarily causes the age-dependent neurodegeneration in AD.

In sporadic AD, soluble tau species are likely key pathologic contributors to neurodegeneration. Tau pathology has classically been defined in terms of aggregation and the presence of insoluble NFTs, but further characterization of tau’s route to aggregation has revealed that tau undergoes multiple stages of PTM development before forming insoluble aggregates [24], and that soluble tau species are likely the toxic entities [2]. However, the ways in which PTMs lead to specific cellular dysfunctions are unknown. Some PTMs may seed the accumulation of other PTMs, some may change tau’s cellular availability, and others may be directly toxic, to list a few possibilities. Hence, considering how PTMs modulate the biochemical properties and localization of soluble tau species through the lens of selective vulnerability is likely to provide new insight into their pathological effects.

## 3. Selective Vulnerability in Sporadic AD

As mentioned above, the progression of sporadic AD tau pathology follows a specific spatial progression in most cases, with the accumulation of NFTs throughout the brain forming the basis of Braak-stage classification for AD pathological diagnosis and directly corresponding to disease severity [9]. Because tau pathology mirrors the spatial progression of neuronal failure in sporadic AD, exploration of selective vulnerability highlights a navigable gap to understanding tau-induced neurodegeneration in AD.

In the classic spatial progression of AD, NFTs first form in the entorhinal cortex (EC) and peripheral cortex [Braak 1], then progress in order through the CA1 region of the hippocampus [Braak 2], the limbic regions [Braak 3], the amygdala, thalamus, and claustrum [Braak 4], the isocortical areas [Braak 5], and ultimately the primary sensory, motor, and visual nexuses [Braak 6] (Figure 2) [11]. Specifically, neurons in layer II of the entorhinal cortex (ECII neurons) are the first to develop NFTs and are therefore considered the most vulnerable or permissive to tau pathology in AD; pyramidal neurons in the neocortex are considered the least permissive [13]. In addition, inhibitory neurons appear more resistant to tau pathology than excitatory neurons, independent of brain region [45,46].

Selective vulnerability is inherently defined as a relative susceptibility to disease, as compared to more resistant neurons. Thus, the genetic factors that confer vulnerability to neuronal subpopulations differ depending on which two populations are compared. This indicates that the most vulnerable neurons likely have multiple, overlapping risk factors that render them most susceptible. For example, recent studies combining single cell and deep sequencing techniques have shown that ECII neurons transcribe fewer mRNAs related to axonogenesis, regulation of synaptic plasticity, microtubule-based trafficking, and autophagy when compared to more resistant cells, such as pyramidal neurons [13], which is consistent with the impact of tau on these processes [13,33,38,49,50,51]. Even though ECII neurons are typically the first to exhibit tau pathology in AD, not all ECII neurons are permissive to AD-induced neurodegeneration [52]. Single nucleus RNA sequencing (snRNA-seq) analysis of cellular subtypes within ECII neurons has revealed that vulnerable ECII subpopulations differentially expressed axon-localized proteins, further indicating that machinery in the axon may be critical for regulating toxic tau development [52]. Other studies have found that excitatory neurons are more susceptible to tau accumulation and thus tau pathology, when compared to inhibitory neurons [46]. In addition, vulnerable ECII neurons exhibited increased expression of RAR-related Orphan Receptor B (RORB), a marker and developmental driver of layer IV neurons [52,53,54,55]. Using RORB as a marker, Leng et al. [52] found that tau preferentially accumulated in RORB + ECII excitatory neurons early in AD development. Though preliminary, these studies provide a foundation to identify selectively vulnerable populations within a restricted cohort of cells and point toward differences in tau dynamics as a potential contributor.

Differences in tau isoform abundance may also play a role in selective vulnerability. Modulation of exon 10 splicing is critical for healthy tau function, as exon 10 encodes the fourth MTBD and therefore regulates the ratio of 3R to 4R tau in the brain [13]. In healthy individuals, 3R and 4R tau are present in roughly equal concentrations [15]. The ratio of 3R/4R tau that is trapped within NFTs is commonly used to distinguish between different tauopathies. For example, the tau aggregates in PiD (pick bodies) are composed of primarily 3R tau, whereas the NFTs in the brains of CBD and PSP cases are comprised predominantly of 4R tau [14,56,57]. NFTs in most cases of FTLD-tau are a mixture of 3R and 4R tau [14]. In the hippocampus of AD-afflicted individuals, 3R and 4R tau are also present in roughly equal concentrations in the NFTs early in the disease process, then 4R predominates in mild-disease and 3R tau predominates in moderate to severe disease [58]. Other studies have shown that an excess of 3R or 4R tau leads to an increase in downstream tau pathology [59]. Specifically, recent experiments using mRNA-trans splicing demonstrated that restoration of the 3R/4R balance in human tau mouse models ameliorated subsequent tau pathology [59]. In addition, multiple studies have shown regional differences in *MAPT* splicing within the brain, particularly with respect to exon 10 splicing [60,61,62]. Thus, pathology induced by an imbalance of 3R/4R tau isoforms across varying tauopathies highlights, at least in part, the importance of tau’s fourth MTBD region in modulating tau-mediated toxicity.

Interestingly, polypyrimidine-tract binding protein (PTB), a regulator of exon 10 splicing, has increased expression in AD-vulnerable ECII neurons compared to resistant neurons [13,63] and may contribute to selective vulnerability by altering the balance of 3R to 4R tau. Specifically, elevated PTB activity increases the abundance of 3R tau in the neuron, which in turn likely increases the relative quantity of soluble tau in the cytosol. The second R region is located within exon 10 and contains one of tau’s two PHF6 domains—a six amino acid motif (VQIVYK) within tau’s inter-repeat region that mediates tau self-association and microtubule affinity [64,65,66]. Because of this, 4R tau is more prone to aggregation [37], yet 3R tau becomes the predominant pathologic tau isoform in the hippocampus during mid to late-stage AD [58]. If 3R tau is less attracted to microtubules and is less prone to aggregation, then the quantity of soluble tau would increase. In vitro, an abundance in 3R tau has been shown to interfere with 4R tau-mediated aggregation [67]. Thus, a potential mechanism through which elevated PTB activity may render certain neurons more vulnerable to AD pathology is by simply increasing the presence of soluble tau in the neuron. Congruent with this idea, mouse studies using repressible mutant (P301L) human tau expression found that tau-induced pathology could be ameliorated or reversed by reducing tau expression, even when NFTs continued to form [2]. In addition, a small molecule inhibitor of Tau phosphorylation has been shown to alleviate P301L-induced motor deficits without altering NFT counts [68]. These studies further indicate that soluble tau species are likely the key pathologic agents in tau pathology [2]. Combining this idea with the observation that RORB + ECII excitatory neurons are more prone to tau accumulation and tau pathology highlights the possibility that increased concentrations of soluble tau may be a critical mediator of the selective vulnerability seen in the neurons most vulnerable to sporadic AD pathology. Differences in PTB and RORB expression within ECII subpopulations permit tau accumulation and alter the species of tau available for PTM. The differences in axonal gene expression for ECII neurons may render neurons more vulnerable to axon-related tau pathology. Although little is known about the topic, differences in gene expression may increase the soluble tau available for toxic modification, and may permit specific tau PTMs to act as primary contributors to the basal tau toxicity seen in early-AD.

## 4. Tau PTM Processivity

Soluble tau generally lacks a rigid three-dimensional structure. Hence, the potential sites of PTM are not occluded by protein folding, leaving the protein scaffold inherently more vulnerable to modification [69]. Many excellent reviews have been written to survey the increasing library of known tau PTMs such as phosphorylation, acetylation, ubiquitylation, and SUMOylation: Alquezar et al. [14] is one recent example. These PTMs in-and-of-themselves can alter the physiochemical properties of tau, perhaps contributing major structural constraints to subsequent folding or interactions with other proteins, and modulating tau solubility, localization, or functionality. Interestingly, cryo-EM studies have demonstrated that the three-dimensional structures of tau filaments (the basic structural components of insoluble aggregates such as NFTs and pick bodies) are unique for individual tauopathies [70,71,72,73]. Since many of these tauopathies are driven by defined mutations in the protein coding sequence, this highlights that the mechanical properties of pre-aggregated tau are also likely different in each tauopathy. This identifies a critical time in tau’s pathological development where structural contributions play a disproportionate role in subsequent toxicity. This evokes the need for more careful consideration of how tau PTMs influence the physiochemical properties of tau.

Advances in the mass-spectrometry-based characterization of tau PTMs have indicated that, like the stereotypical progression of tau pathology through brain regions, PTMs also probabilistically accumulate in a specific order as AD progresses [24]. PTM of tau, particularly phosphorylation, has been well studied using antibody-based approaches [74,75,76]. The longest tau isoform (2N4R) contains 85 potential serine, threonine, or tyrosine phosphorylation sites, with approximately 30 of these sites possessing various levels of occupancy in normal tau proteins [20,77]. The relationship between tau splicing, solubility, and the distribution or progression of PTM is complex, and antibody-based methods do not allow for robust characterization of individual modifications at the single protein level, nor do they allow for stoichiometric analysis of all modifications in a tissue.

Nonetheless, unique mass-spectrometry analyses of tau are now providing detailed insight into the stoichiometric distribution of select tau PTMs as they accumulate in vivo [24]. Using these methods to track tau PTM profiles alongside defined stages of AD progression has demonstrated that discrete PTM “signatures” correlate with specific stages of AD disease severity [24]. In other words, tau PTMs accumulate in a processive fashion, and specific classes of modifications correlate tightly with Braak staging in human post-mortem analyses [24]. Based on these findings, it is enticing to imagine that specific Braak stages are caused by distinct PTM-profiles that accumulate on endogenous tau and alter its conformation and localization. Because soluble tau is a key mediator of AD pathology [2,3,78], identifying early pathological PTM-profiles in soluble tau species could be critical for understanding AD-associated selective vulnerability. We herein refer to the processive, PTM-profile-dependent development of tau into a soluble toxic entity as “tau transformation” (Figure 3).

## 5. Tau Transformation—Early Pathological Changes in Tau Localization, Conformation, and Structure

There is increasing evidence that tau localization may be impacted in AD and contribute to the accumulation of certain PTMs in early tau transformation. Normal tau is mainly localized to the axon, where it enhances microtubule dynamics and supports the formation of long labile domains [27]. In addition, tau can complex with Annexin 2 to link plasma membrane and microtubule interactions [79,80,81], and several studies have shown that tau can localize to the nucleus in both healthy and diseased brain tissue [80,82,83,84], though tau’s role there remains a question. In addition, trace amounts of tau can also be found in the dendrites and may play a role in synaptic plasticity [85,86]. Axonal localization of tau is maintained in part by modulation of tau’s microtubule binding by specific phosphorylation sites, as certain phosphorylated tau (pTau) epitopes have been shown to induce free-diffusion of tau from axons to the rest of the neuron [87]. Furthermore, the axon initiating segment (AIS), an F-actin barrier that maintains neuron polarity, serves as a diffusion barrier and retains higher concentrations of tau in the axon under physiological conditions [37,87,88,89], indicating that axonal localization of tau may be a critical attribute of neuronal health. Once translated, tau can migrate to the axon in multiple ways: it can diffuse through the cytoplasm, diffuse along microtubules, or it can be shuttled directly to the axon via motor proteins [90]. Interestingly, studies have also shown that tau mRNA’s 3′UTR directs it to the axon [91], suggesting that both pre- and post-translational mechanisms might act in concert to deliver and maintain an axonal population of tau. Whether tau that is synthesized in the axon is handled differently from tau that arrives there through other means is unclear, though some clues are emerging. For example, studies performed by Gomes et al. [92] in Drosophila and post-mortem human tissue demonstrated that tau mis-localizes early in AD pathogenesis, and that differential phosphorylation of axonal vs. cytoplasmic tau may initiate tau transformation. In control neurons, tau retained stoichiometrically low levels of phosphorylation, and it localized to the axon. By analyzing the spatial distribution of specific tau species in early, mid, and late-AD entorhinal cortex neurons, they found a differentially increased abundance of pT231 tau in the soma (IC-tau) and pS396/pS404 tau in the axon, synapses, and neuropil (IN-tau) [92]. They confirmed that this specific spatial distribution of tau preceded pre-tangle and NFT formation in all human AD cases, P301L Drosophila, and human wild-type tau Drosophila models. Differential tau phosphorylation was consistently followed by further tau mis-localization and pre-tangle formation throughout the neuron [92]. This finding is consistent with other studies that have demonstrated that tau’s mis-localization from the axon may represent an early event in tau-induced synaptic dysfunction and neurodegeneration [88,92,93,94].

Phosphorylation events early in tau transformation have been shown to reduce tau’s affinity for microtubules, particularly when accumulated in the proline-rich domain (PRD) just upstream of the MTBD [95,96,97]. The PRD plays a regulatory role in microtubule binding [21,98], as its average isoelectric point of >9 is positively charged at physiologic pH and stabilizes binding to the negatively charged microtubules; the PRD also helps to offset the electronegatively enriched MTBD [99]. In general, phosphorylation within the PRD or C-terminus of the MTBD increases the negative charge density and disrupts microtubule association. Specifically, phosphorylation at S214, T231, S235, and S262 have all been shown to decrease microtubule binding affinity [23,40,49,50,100,101]. Gomes et al. [92] reported that spatial segregation of IC-tau (pT231) and IN-tau (pS403/S404) always occurred before AD pathology and preceded pre-tangle formation in 20 separate analyzed human brain regions, indicating that this simple alteration in tau localization may represent the first “irreversible” stage in AD pathology.

In addition to impacting microtubule binding, tau PTMs can also influence its interactions with other binding partners, which may alter the subcellular distribution of the modified tau protein (Figure 3). For example, in Drosophila, bridging integrator 1 (BIN1), the second highest risk factor for sporadic AD in humans [102], has been shown to exclusively traffic tau and keep it within the axon under healthy conditions [92,103,104,105,106]. Phosphorylation within the PRD, and specifically of T231, decreases BIN1′s affinity for tau by interrupting the interaction between the tau PRD and BIN1′s SH3 domain [106]. Loss of BIN1 activity intensifies as sporadic AD progresses [107], likely leading to cytosolic pT231 tau and pS396/S404 accumulation [92]. This single example illustrates how the proper localization of tau can have serious implications for AD development. Loss of control over tau localization through a decrease in factors like BIN1 could represent an initiating step in tau transformation. Once initiated, it is likely that these early phosphorylation events lead to the accumulation of other PTMs on tau. For example, the accumulation of heavily acetylated tau subsequent to phosphorylation has been shown to degrade the AIS [21,88], which may contribute to the loss of axonal tau segregation seen shortly after the appearance of IC and IN-tau in the Gomes model [92]. It is interesting to speculate whether pT231 is pathologic in-and-of-itself, or whether it acts more like a gatekeeper, permitting additional PTMs like acetylation to accumulate on the road to pathogenicity, and whether loss of BIN1 mirrors the progression of selective vulnerability.

A recent study described a method for quantifying tau PTMs via mass-spectrometry (termed “FLEXITau”) to address how their progression occurs and influences disease [108]. Comparing AD-brain tissue to control brain tissue revealed a disease-relevant PTM accumulation pattern that aligns tightly with the model proposed by Gomes et al. [92]; where stoichiometrically high concentrations of pT181 and pT231 were enriched in nearly all clinically and histologically diagnosed cases of AD [24]. This supported the conclusion that phosphorylation accumulation in the PRD signifies the first stages of early AD pathogenesis, whereas heavy acetylation and ubiquitylation were indicators of late-stage disease. Specifically, it seems early phosphorylation at sites like T181 and T231 may initiate more extensive tau alterations, oligomerization, and NFT formation [24,92]. Interestingly, phosphorylation at T231 correlates strongly with phosphorylation at sites in the C-terminus, like S396 [109], which may play a part in C-terminal cleavage of tau [24,37] and tau aggregation [110]. According to the model proposed by Wesseling et al. [24], C-terminal cleavage of tau typically occurs soon after early phosphorylation events in the PRD, followed by tau oligomerization, pre-tangle formation, and NFT accumulation. In addition, phosphorylation at T231 is associated with irreversible aggregate formation in hamster models, whereas aggregates induced by pS199 or pS202/pT205 are readily dissolvable and reversible [111,112]. Consistently, studies have demonstrated that T181 and T231 are critical sites within tau and that accumulation of tau phosphorylated at these sites likely mediate a critical and early step in tau-induced AD pathology [24,113,114,115,116]. Enrichment of phosphorylation in the tau PRD at sites like T181, T217, T231, and S235 correlates strongly with tau’s seeding competency and AD severity [117,118]. Furthermore, accumulation patterns of pT231 follow the same spatial progression as that observed for NFTs in AD [115,119]. Interestingly, concentrations of pT181 tau in the cerebrospinal fluid (CSF), or even in the blood plasma, can be used as a reliable diagnostic marker for AD-prone individuals [120]. These concentrations correlate with disease severity, increasing as symptoms worsen [120]. Thus, specific phosphorylation sites within the PRD, like T181 and T231, may be key-modulating modifications or seeding sites for tau’s transformation into a toxic soluble species.

Importantly, not all phosphorylation events on tau are equal in extent and functional outcome. Highly phosphorylated tau is present in developing neurons and the brains of infants, yet it is not pathologic [121]. In addition, tau is hyper-phosphorylated during normal sleep and hibernation but reverts to physiologically normal PTM profiles shortly after waking [112,122]. Certain PTMs, such as pT205 and pS305, may even be protective by hindering the development of toxic PTM profiles [123,124]. Thus, tau transformation is likely not a discrete event, but rather the result of a slowly changing cellular environment that pushes tau toward a pathologic state.

Early tau transformation may, in part, be managed by PTM competition. For example, different PTMs may compete for the same amino acid in tau, resulting in different functional outcomes. Tau can be phosphorylated and O-GlcNAcylated at the same sites, and O-GlcNAcylation may prevent phosphorylation at key sites [14]; indeed, levels of O-GlcNAc are decreased in AD tissues [125]. Lysine residues may undergo acetylation, but also methylation, ubiquitylation, SUMOylation, or glycosylation [14]. Ubiquitylation can prevent acetylation, particularly within the MTBD [126]. Each of these individual PTMs may impact tau function uniquely, and the context required to impart these modifications is far from being properly characterized. In addition to competing with each other, tau PTMs can also cooperate to induce certain PTM patterns [14]. For example, SUMOylation at K340 enhances phosphorylation at T231 and S262 [127]. While there is undoubtedly systems complexity that makes it difficult to derive first principles, acknowledging this complexity is essential to developing a more nuanced understanding of tau transformation’s role in pathogenicity.

Adding even more complexity to the dynamics of tau PTM are the many observations that kinases, phosphatases, and other upstream enzymes that modulate PTMs are altered in AD and thus contribute to the dynamics of tau transformation. For example, the activity of protein phosphatase 2A (PP2A), the class of heterotrimeric phosphatases that are responsible for ~70% of the protein dephosphorylation events in human brains, is reduced by approximately 50% in AD cases [128]. Truthfully, the extensive repertoire of signal transduction mediators that have been shown to modify tau through their kinase or phosphatase activities makes it difficult to “separate the wheat from the chaff.” Thus, while it is interesting to speculate that differences in GSK3β, AMPK, and PP2A activity, caused by age-induced differences in Akt, mTOR, or ERK signaling [129,130,131], may impart selective vulnerability by increasing concentrations of pathologic tau PTMs in early AD development, such an idea would be significantly strengthened by correlating specific signal transduction cascade activities with known dynamics of selective vulnerability.

In summary, not all tau PTMs, particularly phosphorylation events, are equally pathologic. It is unclear the extent to which most PTMs influence or interact with each other, and it is also unclear how other background factors in the cell control their phosphorylation and pathological influence. Early phosphorylation events (such as pT181 and pT231) may seed a PTM cascade, one that leads to the accumulation of more pathologic PTMs. The process is likely dynamic and likely imparts pathological stress at multiple points in tau transformation. Understanding how these early modifications in tau develop, and ultimately what their prerequisites and impacts are, is key to understanding how tau transformation may create the underlying threat that mediates selective vulnerability.

## 6. Tau Spreading and Seeding

In AD, the mechanism by which pathologic tau spreads from the entorhinal “epicenter” to subsequent brain regions in sporadic AD is still a critical matter of debate. Multiple studies have shown that tau is released from neurons in healthy conditions and that tau is present within the extracellular spaces of the brain [132,133]. Some of this tau is released via exosomes and ectosomes, though most of it is free of surrounding membranes [117,134,135]. Interestingly, recent studies have suggested that tau can directly travel from cell to cell through actin-filament nanotubes that facilitate the exchange of cytoplasmic proteins between adjacent cells [136,137]. In addition, tau can be imported into neurons by bulk endocytosis [138] or actin-dependent micropinocytosis [139,140]. Other studies have shown that tau can propagate through trans-synaptic transmission from neuron to neuron [141], and that tau transport can be mediated by microglia [142,143,144]. Despite many studies on tau transport, the physiological mechanisms behind tau seeding in disease are unclear. One potential mechanism for the spread of tau pathology is that the root-cause is genetic and that neurodegeneration arises from aging-induced, pre-disposed dysfunction; the observed pattern of spatial development would thus be indicative of independent cellular failure in the subsequent regions. This hypothesis requires genetic or proteomic differences between cell populations to be the defining factor of vulnerable neurons. This is the premise for gene-expression studies of selective vulnerability. As discussed above, vulnerable ECII neurons differentially express RORB and PTB, among other factors [13,52]. While differences in PTB activity may influence tau isoforms and alter its transformation into a soluble toxic entity, concrete examples of how these factors influence selective vulnerability are lacking.

Instead, another explanation for the spatial progression of tau pathology is that toxic tau is generated within susceptible cells—likely starting in the excitatory layer II neurons of the entorhinal cortex—and then that toxic tau is released from those cells, taken up by neighboring cells, and propagates the creation of toxic tau as it spreads (Figure 2). A key study demonstrated that viral expression of human tau exclusively in the entorhinal cortex was sufficient to drive tau pathology throughout the brains of mice; tau spreading was higher in aged mice, indicating that older brain tissue is more permissive to tau propagation [145]. In addition, injecting brain extracts rich in P301S tau into healthy mice expressing human tau induces the spread of tau pathology into the regions neighboring the injection site [146]. This indicates that pathologic tau can convert otherwise healthy tau into toxic tau. Thus, the age-dependent release of pathologic tau from “host” cells may be a critical contributor to spatial disease progression. As such, recent studies have started focusing on the “prion-like” seeding potential for toxic tau species as a critical component of tau pathology [147,148].

Tau truncation may be a PTM critical for tau seeding and toxic tau propagation, and recent insights into PTM processivity are highlighting the connection between early phosphorylation events and C-terminal cleavage of tau. Specifically, phosphorylation of tau alters its conformation when soluble, and predominantly occurs within the PRD early in tau transformation. Phosphorylation of the PRD, at sites such as T181, S202, T205, and T231, has been shown to alter tau’s conformation [149,150]. Physiologically, tau can adopt a closed, “paperclip” conformation where the N-terminus and C-terminus fold inward and are stabilized by the mid-region of tau [151]. Phosphorylation at S202 and T205 leads to the thermodynamic dissociation of the N-terminus from the tau mid-region, and phosphorylation at S396 leads to C-terminal dissociation [150], thereby opening one of tau’s native conformations and increasing the availability of the N and C-terminus for further modification [149]. In addition, Wesseling et al. [24] observed that phosphorylation at S403/S404 occurs soon after PRD phosphorylation and is tightly linked to C-terminal cleavage of tau through unknown mechanisms. Congruent with this, classic tau experiments observed that the nucleation core of NFTs was comprised of 12 kDa tau fragments, leading to the hypothesis that C-terminal cleavage is the seeding event for tau oligomerization [152], and stoichiometric mass-spectrometry analysis of seed-competent tau shows increased phosphorylation immediately prior to the C-terminus compared to seed-incompetent tau [24]. Thus, C-terminal cleavage generally occurs downstream of PRD phosphorylation and may even be facilitated by conformational changes induced by early PRD phosphorylation. This provides a clear mechanism through which the accumulation of simple PTMs, like pT231, may contribute to tau truncation. Furthermore, N and C-terminally truncated tau has been shown to induce tau oligomerization and aggregation even in other non-pathologic tau species [24], and these epitopes are exported from neurons in higher abundance than other tau species as well [153,154,155,156]. Finally, truncated tau epitopes can induce mitochondrial stress [157] and activate microglia, inducing inflammation through IL-1β, IL6, and TNF-α signaling [158,159,160], and thereby may exacerbate late-stage disease. These findings highlight the potential for early PTMs in tau transformation as key contributors to tau spreading, in addition to basal toxicity.

As discussed above, increasing evidence is demonstrating that tau’s propagation throughout the brain is a key component of tau toxicity [141,145]. C-terminally or N-terminally cleaved tau may be a critical contributor to tau export and tau toxicity. Regardless of the propagation mechanism, extracellular tau concentrations increase as AD progresses, and concentrations of defined pTau epitopes in CSF can even be used to reliably diagnose disease severity [161,162,163]. Specifically, pT181 and pT217—other PRD phosphorylation sites associated with early tau pathology—see increased concentrations in the CSF as disease progresses [161]. This increase seems to occur independently of neuronal death, indicating that extracellular accumulation of these pTau epitopes is in part mediated by an active export mechanism [161]. Since early PRD phosphorylation likely occurs just upstream of C-terminal cleavage in tau transformation, it is interesting to speculate whether truncation of these diagnostic, extracellular tau epitopes occurs before their export. Both N-terminally and C-terminally cleaved tau are generated when caspase activity is increased, thus toxic tau PTMs that accumulate early in disease may drive tau truncation through induction of neuronal stress and apoptotic machinery [164].

Furthermore, studies have shown that small molecules that block NFT formation do not hinder tauopathy progression, though antibody based anti-NFT therapies do [74,75,165,166]. Many antibody-based approaches also clear extracellular tau [76], further supporting that the rate of disease progression and severity of selective vulnerability are dependent on the spread of toxic, extracellular tau. Thus, the spatial pattern of selective vulnerability in sporadic AD is simultaneously dependent on genetic predisposition and tau propagation.

## 7. When the Levee Breaks—Aging, Proteostatic Clearance, and Failing Quality Control

Proteostatic clearance machinery and tau regulate each other in a cyclical, two-way relationship. Specific tau PTMs can obstruct proteostatic clearance [167], and defects in clearance thereby increase tau abundance and cellular availability, which can further inhibit clearance. Tau which has developed toxic PTMs may be cleared under healthy conditions but begin accumulating when other PTMs impair clearance machinery [168], providing a clear mechanism by which tau PTMs can induce cellular stress and amplify toxic PTM accumulation. In other words, certain tau PTMs may accumulate in AD simply because they hinder their own degradation; others may be detected preferentially in diseased tissue because they are toxic. The accumulation of both classes of tau PTMs may be necessary for disease to manifest, and these complex interactions are at the core of tau toxicity and tau-mediated selective vulnerability. This is namely because quality control and clearance mechanisms are known to deteriorate with age, and modulations in proteostatic clearance directly impact the accumulation of toxic tau [169]. Deficiencies in tau clearance provide two major premises for tau-mediated selective vulnerability: (1) Improper clearance leads to an accumulation of damaged or dysfunctional tau, and this could be a mechanism through which toxic tau species reach pathologic concentrations, and (2) An increase in toxic tau abundance also increases the pathologic tau available for extracellular export and toxic spread (Figure 2).

In part, dysfunction in BCL2-associated athanogene 3 (BAG3), which is a key factor in proteostasis, may contribute to tau accumulation in diseased neurons. Specifically, a study using snRNA-seq analysis of post-mortem human brain tissue found that excitatory neurons—which are more vulnerable to AD pathology than inhibitory neurons, regardless of disease severity or brain region—express more proteins associated with tau aggregation and fewer proteins designed to mitigate aggregative tau stress [46]. The key gene that was found to be expressed at lower levels in excitatory neurons compared to inhibitory neurons was BAG3 [46]. Interestingly, overexpression of BAG3 reduced pTau accumulation in excitatory neurons, and depletion of BAG3 increased pTau levels in inhibitory neurons, which highlights BAG3′s central role in mediating tau pathology in these two different neuronal cell types [46]. Because BAG3 protects against aggregate-induced stress, these findings underscore one mechanism that may confer AD-resistance to inhibitory neurons: increased efficiency in pTau clearance via. BAG3.

In addition to proteostatic stress, PTMs on tau regulate its mechanism of clearance and may compound cellular stress through raw tau accumulation, or through the stoichiometric accumulation of specific tau species that possess toxic PTMs. In healthy tissue, tau is a long-lived protein that is predominantly cleared through cellular autophagy, yet tau turnover becomes more complex as tau pathology progresses [168]. In classical cell culture models, soluble tau species have been proposed to be cleared predominantly through the ubiquitin-proteosome system (UPS), but insoluble or aggregated tau species are cleared through general autophagy [170,171]. Recent studies have demonstrated that tau also accumulates in endosomes and is cleared through endocytic pathways [167,172]. UPS efficiency decreases in AD brain [173], and it has been shown that phosphorylated tau can also directly inhibit the UPS by binding to the 20S proteosome [174,175]. Additionally, autophagy has also been shown to be generally hindered in tauopathies [176,177,178,179]. A study using an immortalized mouse cortical neuronal model found that C-terminal truncation of tau at D421 switched from being predominantly degraded in anon-ubiquitin-dependent manner by the proteasome to general autophagy, suggesting that PTMs alter tau’s mechanism of clearance [180]. Furthermore, a separate study using adenovirus gene-delivery in mice found that tau truncated at D421 develops unique phosphorylation patterns compared to full-length tau; D421 tau also uniquely induced cognitive impairments in aged mice, oligomerized more readily, and caused neurodegeneration in the hippocampus [181]. Studies performed by Vaz-Silva et al. [167] found that tau is frequently trafficked through the entire endolysosomal pathway, binds to endosomal sorting complexes required for transport (ESCRT) machinery in a ubiquitin-dependent manner, and is sorted by Rab35 for efficient movement through endosomes and subsequent degradation by lysosomal fusion. Importantly, they found that not all pTau epitopes were sorted and degraded through the same pathway or with the same efficiency. Namely, pS396/S404 and pS262 tau were predominantly degraded through the ESCRT/Rab35 pathway, but pS202 tau was not [167]. Thus, phosphorylation, ubiquitylation, and C-terminal cleavage events early in tau transformation could be key instigators for tau accumulation.

Inefficiencies and/or failure in protein clearance mechanisms, including that of tau itself, have been clearly linked to AD (for recent reviews, see [12,168,172]) and dysfunctional proteostasis is a general hallmark of age-dependent neurodegenerative diseases [182,183]; the propensity for tau spreading also increases with age [145]. Since age is also the most prevalent risk factor for AD, it stands to reason that the formation of toxic tau and lack of its clearance occur simultaneously and are likely a product of age-related inefficiencies and changes in cellular machinery. For sporadic AD to spatially progress, toxic tau must not only impart a positive pressure for pathogenesis that propagates from cell to cell, but regulatory mechanisms designed to disarm and clear that pathologic tau must also fail, thus rendering vulnerable cells permissive to disease. Certain tau species or PTMs may contribute to dysfunction in proteostatic clearance mechanisms, and this dynamic battle for proteostatic quality control could be another major mechanism behind tau-induced selective vulnerability in sporadic AD.

## 8. Concluding Remarks

The concept of selective vulnerability provides a unique lens for assessing individual aspects of a complex, heterogenous disease like AD. Specifically, the stereotypical temporo-spatial progression of sporadic AD defines a template for identifying molecules that modify disease risk. Utilizing this template, restricted populations of cells can be compared at different stages of disease or alternatively to other cell types with differing susceptibility with the hope of identifying conserved molecular signatures that contribute to (or mitigate) AD.

Recent advances in studying tau processivity have revealed that the appearance of PTMs, including phospho-epitopes, on tau are generally ordered through a stereotypical sequence of additions, and that this order correlates strongly with disease progression and tau’s oligomerization potential [24]. Hence, both the upstream factors that influence tau PTM addition as well as the downstream outputs of selectively modified tau function, including potential PTM-selective binding partners that regulate tau localization, become relevant targets for modifying disease. Within this context, it has become increasingly important to understand how PTMs regulate tau structure, function, localization, and—ultimately—to define how these factors relate to tau’s pathogenicity. Moreover, since it is still unclear whether pathologic tau is cultivated within the most vulnerable neurons and burgeons to subsequent regions, the role that PTMs may play as mediators of tau’s accumulation, movement across membranes, and transfer between cells cannot be ignored.

In summary, this review highlighted three distinct ways in which Tau PTMs may contribute to selective vulnerability on multiple scales. First, tau localization is altered by accumulation of phosphate functional groups in the PRD, like pT181 and pT231 [92]. PRD phosphorylation may be pathologic on its own, but is also likely the initiating, seeding, or gate-keeping modification for tau’s further development into a toxic entity. Second, many PTMs alter tau’s interaction with clearance and degradation machinery, inducing cellular stress and leading to the accumulation of toxic tau entities within individual cells. Third, PRD phosphorylation is tightly associated with a disruption in tau’s paperclip conformation, which may prime tau for N or C-terminal cleavage, thereby increasing its propensity for extracellular transport, and providing a potential PTM-driven mechanism for tau seeding and propagation. In vivo experiments using phosphomimetic model organisms could begin to establish causative relationships between the accumulation of specific PTMs. If phosphorylation events at specific sites are critical for seeding other modifications for tau transformation; differential PTM patterns would develop in aging, AD-relevant models when comparing phosphomimetic strains to wild-type tau controls. Differential PTM accumulation would also likely be accompanied by temporal changes in tau localization and clearance. The identification of selective epitopes would allow for genetic screening to determine which molecular machinery is specifically involved in mediating tau toxicity in vulnerable neurons, as determined by decreased enrichment. These possible experimental paradigms may have the potential to elucidate whether tau PTMs render neuronal cells vulnerable to AD pathology.

Realistically, tau transformation is likely tightly intertwined with other dysfunctions in cellular machinery, where each progressive stage of tau transformation requires permissive changes in cellular or tissue context. Future tau research must keep the complexity of PTM processivity in mind, as specific tau PTMs may be the causal, seeding pathological changes that mediate soluble tau’s toxicity in AD models. Once a thorough library of tau PTMs is established and the pathologic relevance of each is better characterized, therapies that target individual modifications can be developed to mitigate early, seeding pathology associated with selective vulnerability in sporadic AD.

## Figures and Tables

**Figure 1 biology-10-01047-f001:**
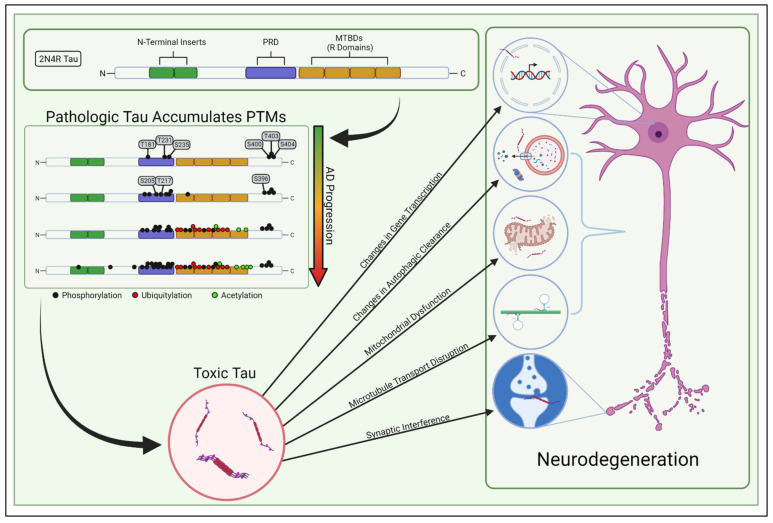
PTM-mediated tau toxicity. Tau isoforms contain up to two N-terminal domains and either 3 or 4 “R domains” (2N4R tau shown, though sporadic AD involves both 3R and 4R tau). The proline-rich domain (PRD) is just upstream of the MTBDs and commonly contains PTMs. The abundance of these PTMs increases stoichiometrically as AD progresses and generates toxic tau species which are likely involved in neurodegeneration. The PTM panel above was inspired by Wesseling et al. [24].

**Figure 2 biology-10-01047-f002:**
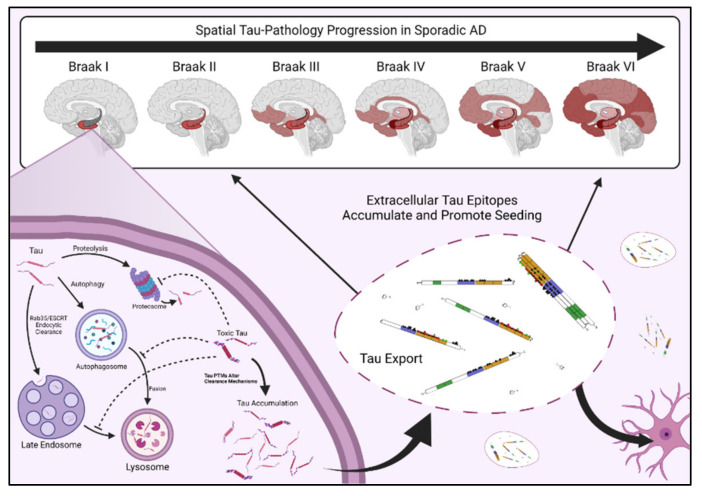
Selective vulnerability and tau accumulation, seeding, and spreading. Tau pathology follows a set track through the brain in most sporadic AD cases [47,48]. Accumulation of NFTs and soluble tau PTM epitopes typically follow the same path as the disease progresses. In vulnerable neurons, tau PTMs can interfere with tau clearance and lead to tau accumulation. Toxic tau PTMs, like tau truncation, are likely to drive tau export from diseased neurons. Though the mechanisms of tau transfer between neural cells are still being debated, increased abundance in extracellular tau is a hallmark of AD, and toxic tau PTM epitopes may drive the spread of pathology from one brain region to the next.

**Figure 3 biology-10-01047-f003:**
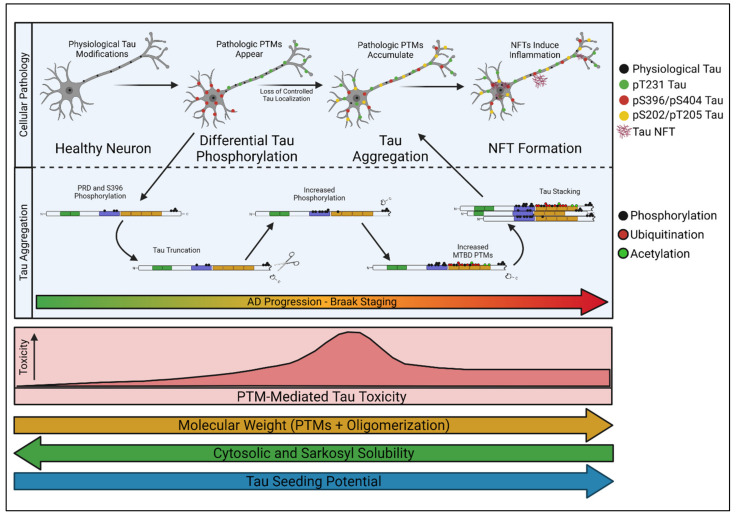
Model for tau transformation. In normal conditions, tau is physiologically modified and is non-toxic. Early in pathology, tau is differentially phosphorylated within the axon and soma of the neuron, leading to defective control over tau localization. These early phosphorylation epitopes, like pT231 and pS396, are associated with further tau modifications and may facilitate subsequent C-terminal cleavage and oligomerization of tau [24]. As disease progresses, tau’s molecular weight and seeding potential generally increase as solubility decreases, but tau’s toxicity is likely greatest when tau species remain soluble.

## Data Availability

Not applicable.

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
