# Peer review of "Tau Post-Translational Modifications: Potentiators of Selective Vulnerability in Sporadic Alzheimer’s Disease"

_biology, 2021, doi:10.3390/biology10101047_

Round 1

Reviewer 1 Report

This is a thorough review of tau pathology in AD, with focus on tau structure and function, Braak staging, and the landscape of post-translational modifications that occur over the course of disease. Strengths of the work include its well designed figures and clear writing style. A weakness is that it does not fully address the issue of selective vulnerability. Suggested minor revisions are summarized below.

Minor points

1. The work would benefit from a clearer definition of selective vulnerability in the introduction and a guide to how each subsection relates to selective vulnerability.

  • With respect to the former, authors should clarify that Braak staging describes a characteristic sequence of brain regions developing tau pathology, which may reflect spreading of a prion-like tau seed through a connectivity chain rather than whether a cell/brain region is intrinsically vulnerable or resistant to tau pathogenesis.

2. Readership would benefit if authors could expand on differences between vulnerable and invulnerable neurons with respect to tau PTM, especially in the contexts they cover. The examples are important because they are among the few that address intrinsic vulnerability/resistance:

  • ECII neurons vs cortical pyramidal neurons (lines 142-144)
  • Excitatory vs inhibitory neurons (lines 145-146)
  • RORB+ vs RORB- neurons (line 167)

3. Readership would benefit if authors could summarize opportunities for experimentation in this area. What approaches could help clarify the role of tau PTM in selective vulnerability? What would they prioritize in this area?

4. Typo line 189: “and” vs “an”

Reviewer 2 Report

The authors present an extensive review article for tau post-translational modifications. The manuscript is well written and the topics they selected seems to be quite appropriate except for few things. Before I can recommend this manuscript to be published, I would like to address a few comments.

1) There are accumulating evidence of plasma phosphorylated tau (and N terminus truncated tau) as a very early biomarker for amyloid beta accumulation. I think including these topics might attract the interest of the readers.

2) The tau proteins in the figures are all 4R taus which could be misleading that 4R tauopathies are relatively minor diseases. 

3) Gene symbols should be in italic (ie MAPT-->MAPT)

Reviewer 3 Report

This is a well-written and comprehensive review on tau post-translational modifications and sporadic Alzheimer's disease. I have only two comments that I think if the authors could address it would strength the work further.

  1. Because this review focuses on SAD, it would be nice to have section on what is known (or not known) about how the molecular pathways/genes identified in newer GWAS studies affect or modify tau/tau pathology. The authors do discuss BIN1, but what is known about other GWAS hits, if anything? In particular, GWAS highlights key molecular pathways such as lipid metabolism, immune function, endocytosis-vesicular trafficking. How might dysfunction in those pathways affect pathogenic tau?
  2. Can the authors comment a bit on the role non-neuronal (glial) cells may play in tau pathogenesis? There is some evidence that tau is expressed in non-neuronal cells, are there the same PTMs? Or, conversely, how are non-neuronal cells affected? There is a bit of information on microglia, but it would be nice if that information was presented in the context of other glial cell types as well.
